# A protocol paper for the MOTION Study—A longitudinal study in a cohort aged 60 years and older to obtain mechanistic knowledge of the role of the gut microbiome during normal healthy ageing in order to develop strategies that will improve lifelong health and wellbeing

**Sarah Phillips[1], Rachel Watt[1], Thomas Atkinson[1], Shelina Rajan[1], Antonietta Hayhoe[1], George M. Savva[1], Michael Hornberger[2], Ben J. L. Burton[3], Janak Saada[4], Melissa Cambell-Kelly[4], Simon Rushbrook[4], Simon R. Carding[1,2]***

**1** Quadram Institute Bioscience, Norwich Research Park, Norwich, United Kingdom, **2** Norwich Medical School, University of East Anglia, Norwich, Norfolk, United Kingdom, **3** James Paget University Hospitals NHS Foundation Trust, Gorleston, Norfolk, United Kingdom, **4** Norfolk and Norwich University Hospital, Norwich, United Kingdom

* Simon.Carding@quadram.ac.uk

## Abstract

### Background

Advances in medicine and public health mean that people are living longer; however, a significant proportion of that increased lifespan is spent in a prolonged state of declining health and wellbeing which places increasing pressure on medical, health and social services. There is a social and economic need to develop strategies to prevent or delay age-related disease and maintain lifelong health. Several studies have suggested links between the gut microbiome and age-related disease, which if confirmed would present a modifiable target for intervention development. The MOTION study aims to determine whether and how changes in the gut microbiome are associated with physical and mental capacity. A comprehensive longitudinal multiparameter study such as this has not been previously undertaken.

### Methods

MOTION is a longitudinal prospective cohort study with a focus on gut health and cognitive function. 360 healthy individuals aged 60 years and older, living in East Anglia, UK will be recruited to the study, stratified into one of three risk groups (cohorts) for developing dementia based on their cognitive function. Participants will attend study appointments every six months over four years, providing stool and blood samples and a health questionnaire. Participants will also undergo physical measurements and cognitive tests at alternating appointments, and undergo Optical Coherence Tomography scans at 3 timepoints. Two subgroups

**Data Availability Statement:** No datasets were generated or analysed during the current study. All relevant data from this study will be made available upon study completion.

**Funding:** The study was funded by the BBSRC through an Institute Strategic Programme (ISP) award to the QIB Gut Health and Food Safety Programme (BB/R012490/1), and its constituent projects BBS/E/F/000PR10353 and BBS/E/F/000PR10356. The study is adopted into the NIHR CRN Central Portfolio Management System (CPMS, add study number and speciality) portfolio which provides additional support in terms of hospital infrastructure and staff support. Dr George Savva is funded through the BBSRC Core Capability Grant BB/CCG1860/1 at the Quadram Institute Bioscience. The Achiever Medical Laboratory Information Management System was procured using the BBSRC Capital Grant Award for the enhancement of the NRP Biorepository. The funders had no role in study design, data collection and analysis, decision to publish, or preparation of the manuscript.

**Competing interests:** The authors have declared that no competing interests exist.

**Abbreviations:** BBSRC, Biotechnology and Biological Sciences Research Council; BCSP, Bowel Cancer Screening Programme; BPQ, Body Perceptions Questionnaire; CBI-R, Cambridge Behavioural Inventory Revised; CCI, Cognitive Change Index; CI, Chief Investigator; eFI, electronic frailty index; GCP, Good Clinical Practice; GP, General Practitioner; HRA, Health Research Authority; IHMS, International Human Microbiome Standards; IVOC, in vitro organ culture; MCI, Mild Cognitive Impairment; Mini-ACE, Mini Addenbrooke's Cognitive Examination; MMSE, Mini-Mental State Examination; MRI, Magnetic resonance imaging; NNUH, Norfolk and Norwich University Hospital; NRP, Norwich Research Park; NSFT, The Norfolk and Suffolk NHS Foundation Trust; OCT, Optical Coherence Tomography; OCT (A), OCT angiography; PHQ9, Patient Health Questionnaire; PIC, Patient Identification Centres; PIS, participant information sheet; QI, Quadram Institute; QIB, Quadram Institute Bioscience; QICRF, Quadram Institute Clinical Research Facility; RCFT, Rey Complex Figure Test; REC, Research Ethics Committee; RNFL, retinal nerve fiber layer; TMT, Trail Making Test.

of participants in the study will provide colonic tissue biopsies (n = ≥30 from each cohort), and brain imaging (n = 30) at two timepoints.

## Discussion

This study will provide new insights into the gut-(microbiota)-brain axis and the relationship between age-associated changes in gut microbe populations and cognitive health. Such insights could help develop new microbe-based strategies to improve lifelong health and wellbeing.

## Trial registration

This study is registered in the ClinicalTrials.gov Database with ID: NCT04199195 Registered: May 14, 2019.

## 1 Introduction

Advances in medicine and public health mean that people are living longer; alongside demographic drivers this results in an ageing population. However, a significant proportion of that increased lifespan is spent in a prolonged state of declining health and wellbeing which places increasing pressure on medical, health and social services [1]. There is a pressing social and economic need for research to promote health and independence into old age. The development of strategies to maintain lifelong health is a major goal over the coming decades.

In older individuals an array of complex and characteristic clinical changes that includes a basal proinflammatory state (so called "inflammaging"), can directly interface with gut microbes of older adults that can enhance susceptibility to diseases accompanying aging, and reduce healthy lifespan [2, 3] which makes the gut microbiome a promising target for strategies that contribute to the health status of at risk and vulnerable individuals. Associations exist between the intestinal microbiota and a variety of age-related conditions, including frailty, *Clostridium difficile* colitis, vulvovaginal atrophy, colorectal carcinoma, and atherosclerotic disease. These links are however, principally seen in cross-sectional studies and it is not known whether the gut microbiome contributes to the development of age related disease [4].

Age is a major risk factor for the development of cognitive dysfunction with dementia being one of the most common disorders linked to ageing [5]. Dementia affects an estimated 47 million people worldwide and is projected to affect over 131 million people by 2050 [6]. Cognitive function often declines with age, when cognitive decline is greater than expected for an individual's age and education level, but doesn't interfere notably with activities of daily life, this is referred to as mild cognitive impairment (MCI). Up to 50% of people with MCI will go on to develop dementia within 5 years [7].

The development of new treatments to prevent dementia is hindered by a lack of predictive biomarkers. The gut microbiome and microbiome-derived metabolites have been identified as promising areas for identifying novel biomarkers of cognitive decline with age [8].

The fact that structural changes within the gut microbiome (dysbiosis) occur during ageing is a consistent finding across different studies [9–11]. However, the scale and nature of the changes observed vary considerably between studies, and among individuals within the same study. Large longitudinal studies with repeated sampling of the gut microbiome would provide a clearer picture of how the gut microbiome changes during ageing, and whether it is a

contributing factor to declining health in old age and/or in the development of neurodegenerative disorders and dementia.

Retinal morphometry also has potential for use as a biomarker for ageing and the risk of dementia. Optical Coherence Tomography (OCT) is a rapid, non-invasive imaging tool that can produce 3- dimensional cross-sectional images of the retina, and permits precise and accurate measurement of the thickness of individual retinal components [12]. In a recent multicentre study of individuals aged between 40 and 69 years of age, OCT imaging revealed that a thinner retinal nerve fiber layer (RNFL) was associated with worse cognitive function in individuals with no neurodegenerative disease and with subsequent cognitive decline [13]. This makes a strong case for regarding retinal anatomical measures as a potentially useful screening marker to identify those at risk of developing dementia.

The studies carried out to date highlight the need to develop a more holistic and integrated understanding of human ageing, in particular how (or whether) changes in the gut are related to changes occurring elsewhere in the body. The MOTION study will obtain a clearer picture of how the gut microbiome changes during ageing in a cohort over the age of 60 years, and how these changes relate to changes in frailty and health. A comprehensive battery of cognitive tests, along with MRI scanning, OCT and physical frailty assessments combined with blood sample collection and six-monthly measurement of the gut microbiome will enable a more detailed investigation of these links than has previously been possible. Such a comprehensive longitudinal multiparameter-based in a well defined cohort of ageing and otherwise healthy individuals is novel and has not been previously undertaken.

## 2 Methods

### 2.1 Aim of the study

The MOTION study will improve our understanding of the links between gut microbial populations and declining health in old age, with a focus on cognitive impairment.

### 2.2 Objectives

The primary objective of this study is to describe and define the composition of the gut microbiome during ageing in a cohort of individuals (60 years of age and older) without existing serious health conditions.

Secondary objectives include;

- the establishment of data and a sample repository to facilitate future research into ageing;

- to estimate how gut microbe populations affect aspects of declining health including the gut, brain, immune and eye function and frailty.

### 2.3 Study design

The MOTION Study is a longitudinal prospective study of 360 participants aged 60 years and older, stratified into three equally sized groups based on their cognitive function (Fig 1).

Participants with any severe medical conditions that are known or are suspected to significantly affect either the gut microbiome or cognitive function at baseline will not be included. For full inclusion/exclusion criteria, refer to Table 1.

Recruitment began in October 2019 and will continue until October 2022. Participants will be recruited through three different mechanisms (see section 2.4) that together aim to identify participants from across the cognitive spectrum from healthy to those with mild cognitive

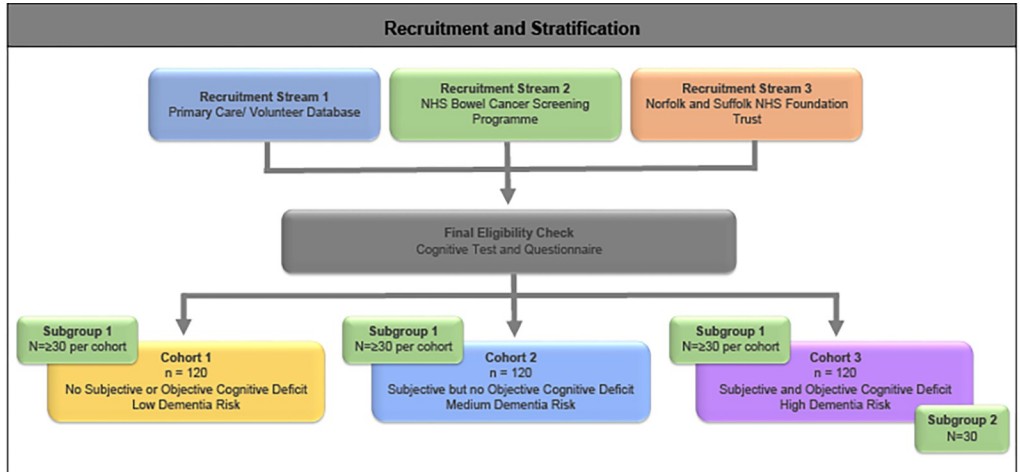

**Fig 1. MOTION study design, schedule of enrolment, interventions and assessments.**

impairment, while also enabling the collection of intestinal biopsies from a significant number through the course of their routine clinical care, via the UK Bowel Cancer Screening Programme (BCSP).

All participants undergo the Mini Addenbrookes Cognitive Examination (mini-ACE) and the Cognitive Change Index (CCI) [14, 15] as a final eligibility check and for stratification. Based on these scores each participant will be stratified into one of three cohorts, which will each stop recruiting once the required sample size of 120 has been reached.

- **Cohort 1** –No subjective and no objective cognition deficit (low risk of developing dementia). Mini-ACE score of ≥26/30 and CCI score of <16 (from the first 12 questions) at consent appointment.

- **Cohort 2** –Subjective but no objective cognitive deficits (medium risk of developing dementia). Mini-ACE score of ≥26/30 and CCI score of ≥16 (from the first 12 questions) at consent appointment.

**Table 1. MOTION study inclusion and exclusion criteria.**

| Inclusion Criteria: |
|---|
| Be at least 60 years old |
| Be able to understand the study and provide informed consent |
| Be able to provide a stool sample within 24 hours of each study visit |
| Be willing to undergo blood tests at each visit |
| Be able to complete cognitive tests / questionnaires and be familiar with using an ipad/tablet (support will be available if necessary). |

| Exclusion Criteria | |
|---|---|
| **Participants will be withdrawn if these develop during the study** | are currently taking part in an interventional study |
| | are living with or related to any member of the research team |
| | have a diagnosis of; dementia, Parkinson's disease, Alzheimer's disease, Creutzfeldt-Jakob disease (CJD), Picks disease, schizophrenia, bipolar disorder, obsessive compulsive disorder, epilepsy |
| | have had a stroke |
| | have current, untreated clinical depression |
| | have an irreversible brain injury |
| **Participants may need to be withdrawn if these develop during the study** | take more than a daily dose of probiotics |
| | have cancer currently, or history of cancer in the last 5 years, except for squamous or basal cell carcinomas of the skin that have been medically managed by local excision |
| | have a long-standing gastrointestinal or liver function abnormality requiring on-going medical management or medication |
| | have made any major changes to your diet in the last month (for example changed to a vegan/ vegetarian diet or stopped eating red meat) |
| | have a history of any liver problems, for example hepatitis B or hepatitis C |
| | have a history of alcohol, drug or substance abuse |
| | have had major surgery of the gastrointestinal tract, apart from gall bladder or appendix removal in the past five years |
| | have had any major bowel surgery at any time |
| | have a history of an inflamed bowel, for example ulcerative colitis, Crohn's disease or diverticulitis |
| | have persistent, infectious gastroenteritis, colitis or gastritis; persistent or long-term diarrhoea of unknown cause; *Clostridium difficile* infection (recurrent) or *Helicobacter pylori* infection (untreated) |
| | suffer with constipation |
| | regularly use laxatives |
| | has an electronic medical implant (e.g. a pacemaker) |

- **Cohort 3** –Subjective and objective cognitive deficits (high risk of developing dementia). Mini-ACE score of <26/30 and CCI score of ≥16 (from the first 12 questions) at consent appointment.

It is anticipated that all three cohorts may be populated from more than one recruitment stream.

At least 30 participants from each cohort will be recruited from recruitment stream 2 that will be due to have a colonoscopy as part of the BCSP. These participants will form subgroup 1. Participants in this subgroup will provide colon biopsy samples if there are routine biopsies being collected as a part of their routine care colonoscopy.

30 participants from cohort 3 (high-risk group) will form subgroup 2 and will undergo Magnetic Resonance Imaging (MRI)-based brain imaging on two occasions (the first within 6 weeks of the baseline visit appointment and the second up to 6 weeks prior to the final study appointment) at the Norfolk and Norwich University Hospital NHS Foundation Trust (NNUH). Each consecutive participant who is enrolled into cohort 3 will be offered the brain MRI scan. If the participant declines this or is not medically able to have the scan, it will be offered to the next consecutive participant until the required number of 30 is achieved. Imaging protocols will adopt the same design and concept as the Alzheimer's Disease Neuroimaging Initiative 2 (ADNI2) protocol (ClinicalTrials.gov Identifier: NCT01231971) to measure the progression of MCI (1h total scanning time) [16].

Participants will complete questionnaires and assessments as well as provide biological samples at approximately 12 appointments (additional appointments will be undertaken if the participant is a part of either sub-groups). See Fig 1 for details.

## 2.4 Recruitment

Three separate recruitment streams will be used to recruit 360 eligible participants aged 60 years and above living in East Anglia, UK. Written informed consent will be obtained following Good Clinical Practice (GCP) guidelines by trained members of the research team. Following consent, all participants will be assigned a unique MOTION study participant number which will provide pseudonymisation for the participant throughout the study. It will be clearly stated that the participant is free to withdraw from the study at any time for any reason without prejudice to future care, without affecting their legal rights, and with no obligation to give the reason for withdrawal.

**2.4.1 Stream 1 primary care.** Primary Care sites across Norfolk will act as Patient Identification Centres (PIC) sites to identify suitable participants by searching their electronic patient record against the study inclusion and exclusion criteria. The PIC site will send an introductory pack to the potential participant, who will respond to the letter or contact the research team by phone /email if they are interested in participating in the study.

**2.4.2 Stream 2 BCSP.** In England, all patients aged 60 to 74 registered with a GP are automatically sent a faecal immunochemical test (FIT) kit every 2 years to detect faecal occult blood which could indicate abnormalities in the bowel which may have the potential to develop into cancer. Patients with a positive FIT result are referred for further investigation by colonoscopy. Patients invited for a colonoscopy are contacted by mail in advance of their appointment scheduled as part of their routine care. In addition to this mail out, the BCSP will send these patients within Norfolk an introductory pack for the MOTION study. If the patient is interested in participating in the study, they are asked to respond to the letter or contact the research team by phone /email. Stream 2 will potentially identify participants for cohorts 1, 2 & 3.

**2.4.3 Stream 3 NSFT.** The Norfolk and Suffolk NHS Foundation Trust (NSFT) Memory Assessment Services receive 200 GP referrals each week in Norwich, South Norwich, North Norwich/Norfolk, Great Yarmouth and Lowestoft. When referred patients are seen for a memory assessment by the Community Teams for each area, they are asked to sign a Consent to Research form which allows the NSFT research team to approach them for potential research studies. The NSFT research team will approach those patients with a diagnosis of MCI that have completed this form and will provide the patient with an introductory pack to the study. Patients who are interested in participating in the study will be asked to respond to the letter or contact the research team by phone /email. Stream 3 will mainly identify participants suitable for cohort 3.

**2.4.4 Additional recruitment methods.** Participants may also be identified by expressing an interest on the Quadram Institute's website using the online form provided for the study. Additionally, any individuals who have signed up to the Quadram Institute Bioscience Participant Database who meet the eligibility criteria, will be sent the study information pack and asked to respond to the letter, or contact the research team by phone/email if they are interested in participating in the study.

## 2.5 Sample biobanking at NRP biorepository

A key aspect of the MOTION study involves creating biobank of samples and data which can be used in future ethically approved research. To enable this, and following Informed Consent for study participation, all participants will be asked if they would be willing for their samples to be stored anonymously at the Norwich Research Park (NRP) Biorepository during and after the end of the study for long term storage. The NRP Biorepository is a tissue bank licensed by the UK's Human Tissue Authority (HTA) with appropriate ethics approval has appropriate ethics approval which was granted by the NHS Health Research Authority (HRA), East of England-Cambridge East Research Ethics Committee (REC). The MOTION study is creating a biobank of material and data that can be used in future research within the Norwich Research Park, nationally and internationally. Such studies may include genetic studies, animal research, or commercial studies with explicit consent of the participants.

Participants consenting to the long-term storage of their samples at the NRP Biorepository are given the option to provide explicit consent for their samples to potentially be used in future ethically approved animal, cloning, or commercial studies, however, these aspects are optional and not required for samples to be stored long term in the NRP Biorepository. Data will be recorded by returning a copy of the consent form to the Biorepository. The participant will be asked to read the current version of the NRP Biorepository Information Sheet and to sign this consent if they wish to participate in this aspect of the study. If the participant declines long term storage at the NRP Biorepository, all samples at the end of the study will be destroyed appropriately in accordance with the Human Tissue Act requirements.

## 2.6 Data and sample collection

At study visits, participants will provide biological samples and complete questionnaires and cognitive tests according to the schedule described in Fig 1.

Research participants that are successfully recruited onto the study will be assigned a unique code number. All data and biological biological samples collected as part of the study will pseudonymised. The file linking the participant to the unique code will be kept on a secure database and a paper copy held in a lockable filing cabinet or cupboard. Only anonymised individual-level data will be shared within study team members (QIB and NHS investigators). Data will be collected by trained researchers only.

Copies of all test results including any incidental findings will be sent to the participants' GP.

Samples (stool, tissue biopsies and blood) provided to QIB will be processed immediately, or aliquoted and stored in -80°C freezers in QIB, or the NRP Biorepository for long term storage (if the participant has provided consent). Sample management at QIB and the NRP biorepository will be enabled by use of a locally acquired Laboratory Information Management System.

Direct access will be granted to authorised representatives from the Sponsor and host institution for monitoring and/or audit of the study to ensure compliance with regulations.

A data monitoring committee is not needed because the trial is not testing a drug or device and any safety concerns associated with the trial have been reviewed by the ethics committee.

### 2.6.1 Stool samples

Stool samples are collected by participants within 24 hours of their booked study visits at the QICRF to examine the microbiome content. Participants will be provided with a stool sample collection pack before their appointments containing everything they need to collect their samples and instructions on how to collect it. The sample collection pack contains an anaerobic gas producing sachet and icepacks to maintain integrity and viability of the sample until it is brought to QIB. The collection protocol has been developed according to International Human Microbiome Standards (IHMS) guidelines [17].

Upon receipt at QIB, the sample will be logged, dispersed into a homogenous suspension and aliquoted (~0.5g each). Some aliquots may be used immediately upon receipt for determination of water content and for the quantification and extraction of microbes (bacteria, viruses, fungi, archaea), for use, for example, in faecal microbiome transplantation studies, for extraction of faecal water containing metabolites and other microbial products, and/or DNA/RNA. The remaining aliquots will be stored frozen with or without the addition of preservatives (e.g. glycerol) for future ethically approved studies.

Microbiome metagenome analysis will include using high-throughput DNA/RNA sequencing methodologies (e.g. Illumina Mi/HiSeq) according to IHMS guidelines for sequencing and data analysis and specific protocols and bioinformatics pipelines established at QIB that have been developed and used in previous clinical microbiome studies. This includes:

- Optimised DNA/RNA extraction protocols, and library preparation for bacterial, viral, fungal and/or archaea species and taxonomic identification.

- A subset of samples (selected based on clinical metadata and microbial profiles) will be analysed for host (immune/metabolite) molecules.

- Next-generation-sequencing-based metagenomic analysis of faecal microbiome will be coupled with culturing methodologies (single and complex) to identify new strains of health-promoting bacteria and potential pathogens, with corresponding whole genome sequencing performed.

- Microbial analysis will be correlated with health outcomes by referring to detailed clinical information (hospital and GP notes).

**2.6.2 Blood samples.** Blood samples will be collected at each study visit appointment at QI-CRF.

At baseline and study visits 2,4,6 and 8, two 10mL blood samples will be taken in EDTA blood collection tubes and two 10mL blood samples will be taken in Serum Separation Tubes. Samples will be processed and stored in QIB laboratories. Analyses will include, for example, assays to characterise immunoglobulins and antibodies; metabolites; inflammatory mediators (e.g. cytokines); and the presence of microbes and microbial products using established protocols within QIB.

At study visits 1, 3, 5, and 7, 15mL of blood will be used to determine total red cell counts, haemoglobin concentration and for assessment/quantitation of key biochemical markers of organ function and health (e.g. bone, liver, heart, kidney and thyroid). All of these have been identified as significant covariates for microbiome composition [18].

**2.6.3 Physical measurement collection.** Height, body composition measurements, blood pressure and hand grip strength measurements will be taken by a trained member of staff at the QICRF at study visits 2, 4, 6 & 8. Data will be recorded on the study case report form.

**2.6.4 Health questionnaire.** At the baseline appointment, participants will be asked to complete a health questionnaire which asks questions related to demographics, lifestyle, behaviour and hygiene. This questionnaire has been used in previous longitudinal studies of the human gut microbiota and identifies key environmental factors that are known to impact on the structure and function of gut microbes [19].

At subsequent appointments, participants will be asked to complete a shorter version of this questionnaire to collect information about changes in health status, medication use and any other changes in the routine of the participants at the time of producing follow up samples.

**2.6.5 Cognitive tests and associated questionnaires.** The cognitive tests and associated questionnaires used in the MOTION study are:

- **Mini-Addenbrooke's Cognitive Examination (Mini-ACE)** [14]- a short standard cognitive screening test well validated and used in dementia screening.

- **Cognitive Change Index (CCI)** [15]—a short validated self-completed measure of the participant's own perception of their cognitive decline.

- **Patient Health Questionnaire (PHQ9)** [20]—brief self-report instrument for screening the severity of depression symptoms in the participant.

- **Supermarket Test** [21]—a computer- and tablet-based assessment of spatial orientation using an ecological supermarket environment.

- **Sea Hero Quest** [22]—novel computer, tablet, or virtual reality measure of spatial navigation within the context of a video game.

- **Cambridge Behavioural Inventory Revised (CBI-R)** [23]—a self-rated measure of changes in the participant's behaviour. These domains include memory and orientation, everyday skills, self-care, abnormal behaviour, mood, beliefs, eating habits, sleep, stereotypic and motor behaviours and motivation and frequency of these behaviours.

- **Body Perceptions Questionnaire (BPQ)** [24]—a self-report questionnaire assessing the participant's degree of interception ability.

- **Rey Complex Figure Test (RCFT)** [25]—a short measure of visual memory and visuospatial constructional ability which includes a 3-minute delayed recall.

- **Trail Making Test (TMT)** [26]—a short test of processing speed, attention and set-switching.

Participants will be asked to complete these at their Baseline visit and study visits 1, 3, 5, 7.

The mini-ACE will give a general overview of cognitive intactness in participants via an objective test. This is complemented by the CCI which allows participants themselves to report whether they have any concerns in regard to their cognition.

In addition, more specialised cognitive tests, targeted towards the first changes commonly seen in dementia, will be employed to measure specific cognitive domains. For spatial navigation, which is currently emerging as one of the earliest cognitive sign of dementia disease pathological processes, we will use the Supermarket and Sea Hero Quest tests. The RCFT allows a stricter assessment of episodic memory in participants, which is currently still seen as the gold standard of dementia cognitive changes. Finally, the TMT allows to detect more slowing down of response and problems switching tasks, which is commonly found in more vascular brain changes and vascular dementia.

These tests will be complemented by 3 additional questionnaires, PHQ-9 allows establishment of depressive symptoms which can impact on cognitive performance, CBI-R allows establishment of any behavioural/neuropsychiatric symptoms which can impact on cognition; finally, BPQ is a questionnaire which taps into introception, i.e. how well people are aware of their bodily functions. The BPQ will be therefore important to how people are aware of their gut and has been suggested to be a cognitive surrogate marker of gut-brain interaction levels.

**2.6.6 Retinal morphology OCT/OCT(A) scans.** There is a growing literature on the use of Optical Coherence Tomography (OCT) scanning to measure both the Retinal Nerve Fibre Layer (RNFL) and the Inner Ganglion Cell Layer (iGCL) of the retina and correlating these measurements with the risk of dementia [13, 27]. Embryologically the eye is an out pouching from the brain and contains neural tissue. It is possible that if the brain is undergoing a degenerative atrophy as seen in dementia then there might also be measurable changes in the retina, particularly the retinal nerve fibres of the ganglion cells. This might mean that OCT of the RNFL or iGCL becomes a useful non-invasive way of detecting those at higher risk of dementia, or quantitatively measuring the response to novel treatment strategies.

The OCT scan will be performed following the Baseline Visit, around Study Visit 4 and prior to Study Visit 8 in all participants.

We will use the newer technique of Optical Coherence Tomography Angiography (OCT A) at the Baseline Visit & Visit 8 in subgroup 1 (the same 90 participants– 30 from each cohort who undergo a colonoscopy) which, in addition to the OCT scan, measures blood flow in the retina, again in a non-invasive manner. We wish to get OCT (A) scans on the participants in whom we have the most information about the gut microbiome and this will be those in Subgroup 1.

There is minimal data on correlating OCT (A) findings with dementia, or indeed how to quantify and analyse the large amount of data that OCT (A) scans generate [28]. However, the MOTION study is unique in collecting OCT(A) data in ageing participants with and without MCI which will be examined by an Ophthalmologist who is blinded to participants cognitive and clinical data.to see if there is any change in the vascular pattern in participants with MCI (cohort 3), looking at both the retinal and choroidal blood supply, measuring when possible the vascular density and looking for other changes such as enlargement of the foveal avascular zone which might reflect progressive microvascular changes.

*2.6.6.1 Outcome measures.* Outcomes will be extracted from OCT(A) data by an Ophthalmologist who is blinded to participants cognitive and clinical data. Peripapillary RNFL in superior, inferior, temporal and nasal quadrants and mean quadrants will be measured. Macular thickness will be recorded form the OCT and inner ganglion cell layer thickness will be recorded by outer, inner and foveal segments.

For Ishihara test score, choroidal thickness will be recorded from the OCT. OCT (A) data will be analysed for area of Foveal Avascular Zone (FAZ).

Data reported from the right eye unless the OCT grader reports segmentation failure. In that case, data from the other eye will be used if no segmentation failure is recorded for that eye.

Participants with ophthalmic conditions that would interfere with the OCT measurements will be excluded from analysis. This would include diabetic retinopathy, retinal vein occlusion, uveitis and glaucoma.

**2.6.7 MRI scans.** Participants undergoing the MRI scans will attend an MRI scan within 6 weeks of their baseline visit, and again within 6 weeks of the final study visit (study visit 8).

The 60-minute scan will capture both structural and functional, 3D T1-weighted, sequences.

**2.6.8. Colon biopsies.** Microbe populations that are specifically and intimately associated with the lining of the gut wall are ideally situated to influence the host and intestinal cells, but they can be underrepresented or absent in stool samples. For this reason, colon tissue samples are important to: 1) fully identify the microbes that are in most intimate contact with the host 2) identify and analyse how the cells within the tissue respond and react to these microbes and their products.

For colonoscopy biopsies, 6–8 pinch biopsies each measuring 3mm$^3$ will be taken from the large bowel during the BCSP colonoscopy. These biopsies are taken to examine the microbiome that is adhering directly to the mucosal lining of the colon.

Fresh tissue will be immediately used for organ or cell culture systems, or for cell isolation and analysis. Other immediate means of processing tissue include those for (immuno) histology; isolating cells (e.g. immune and epithelial cells) for detailed characterisation (e.g. repertoire and subset identification and functional properties); proteins isolation/quantification (e.g. cytokines and other inflammatory or immunoregulatory proteins); and/or nucleic acids (DNA/RNA) extraction and microbial/host cell gene expression analysis.

For long term storage, biopsy samples will be preserved by, for example, snap freezing in liquid nitrogen (particularly useful for preserving RNA) and/or immersion in formalin (ideal for preserving tissue histology).

**2.6.9 Electronic frailty index data.** The electronic frailty index (eFI) helps identify and predict adverse outcomes for older patients in Primary Care. The eFI is made up of 36 deficits comprising around 2,000 Read codes and is a 'cumulative deficit' model, which measures frailty on the basis of the accumulation of a range of deficits, which can be clinical signs (e.g. tremor), symptoms (e.g. vision problems), diseases, disabilities and abnormal test values [29]. Many of these deficits are associated with age-associated changes in gut microbe populations. Data will be requested from the participant's GP retrospectively at the end of the study. Data will be collected for a single timepoint at the end of the study.

## 2.7 COVID-19 mitigation strategy

To protect the safety of participants and the study team during the coronavirus pandemic, we will:

- not ask participants who are advised to shield to attend research appointments.

- give participants the option to decline to attend their research appointments if they are worried about associated risks of Coronavirus.

- contact participants before scheduled appointments at QI CRF to check if participants (or those that they live with) have displayed any symptoms within the last 14 days, are self-isolating as a result of the NHS test and trace scheme, have tested positive for coronavirus or are awaiting test results. If any of these apply their appointment will be rescheduled.

- schedule appointments to minimise contact with other participants and staff members.

- maintain safe social distancing practices where possible.

- disinfect surfaces that participants are likely to come into contact within our appointment rooms before and after appointments.

- Provide contact details of participants who have attended appointments to the NHS test and trace programme if requested for contact tracing.

## 2.8 Stastics and analysis

Indicative outlines of key questions to address the main study objectives are described below. Details will vary depending on the nature of the data collected, methodological developments in the rapidly evolving area of microbiome analysis and emerging scientific literature throughout the study:

**2.8.1 Interim analysis/ monitoring.**   Interim reports will focus on monitoring the progress of the study against the main objectives to inform any necessary changes in protocol. These will include, as relevant at each stage of the study:

- Monitoring the rate of eligible participants from each stream,

- Participation rate of GPs

- Participation rate among different patient groups

- Rate of accruals from each stream

- Rate of accruals into each cohort

- Proportion consenting to each element of the study

- Demographic and clinical characteristics of the cohorts

- The completeness and quality of data and samples being received

- Patient's adherence to the sample collection protocols

- Response rate to sub-studies

- Attrition rate across different patient groups in each wave

- Rate of adverse events

**2.8.2 Cross-sectional analyses (following completion of baseline assessments).**   Following the completion of baseline data collection, we will have data on the composition of the gut microbiome and associated clinical, demographic and lifestyle data on all participants. This can be used to address the following questions:

1. What is the association between age and the composition of the gut microbiome, and what is the variation in gut microbiome composition and function among middle aged and older people in Norfolk?

2. How do differences in lifestyle between age-groups account for these differences?

3. Is the microbiome associated with cognitive function and associated symptoms, physical function or other markers of successful ageing?

4. Can age or lifestyle differences account for any associations identified in 3 above, or

5. Does the gut microbiome mediate the effects of age and lifestyle on health?

**2.8.3 Longitudinal analyses (after successive waves and following completion of all data collection).**   Detailed analyses plans will be prepared following data collection but ahead of any analysis after each visit, building on the cross-sectional findings, but in brief planned analyses will address the following questions:

6. What is the intra-individual variation and trajectory of different components of the gut microbiome with age?

7. How does the baseline composition of the gut microbiome predict future changes in clinical outcomes? E.g., among people with mild cognitive impairment, can analysis of the gut microbiome be used to predict future cognitive change.

8. How do changes in the gut microbiome correlate with changes in clinical outcomes?

**2.8.4 Numbers of participants.** The study will recruit 360 participants. Over four years we anticipate that roughly 20% of participants will die or drop out of the study, reflecting rates seen in national cohort studies in similar populations over similar periods of time, hence 300 participants would be available for analysis at the end of the study.

A primary aim of this study is to estimate the link between microbiome composition and longitudinal change in clinical outcomes. A simulation study was conducted to estimate power for this objective, using as inputs the dynamics of cognitive change estimated from The Irish Longitudinal Study on Ageing [30]. There are many unknowns with respect to the prevalence of the microbiome features that might be associated with cognitive decline, the size of their effects, and variation in outcome measures. There is no prior data on our primary cognitive outcome measure (Mini- ACE), and so a generic measure of cognitive function (Mini-Mental State Examination, MMSE) was used, which is likely to have similar measurement properties. The standard deviation in MMSE change over four years was 2.2 points in a dataset with similar characteristics to the planned MOTION study. The smallest clinically meaningful difference in MMSE score has been reported as 1.4 [31], although much smaller effects are likely to be of scientific or public health importance. The power to detect effects equivalent to 1.5, 1.0, 0.8 and 0.5 MMSE points (corresponding to standardised effect sizes of 0.68 standard deviations to 0.22 standard deviations) over 4 years were estimated by simulation (at $p < 0.05$), assuming prevalence of the specific microbial feature causing the effect is between 5% and 50%. A mixed effects model was used, with five evenly spaced time points, fixed effects of baseline microbiome and its interaction with time, and random intercept and slope accounting for between participant differences in cognitive change. The power quoted was the power to detect the interaction between the baseline microbiome and time on cognitive change, assuming this effect was constant over the study period. These are shown in Table 2, showing that under reasonable assumptions the MOTION study will have enough power to detect moderate effects of relatively prevalent features, and large effects of rare features.

The simulation was conducted using the simr package in R statistical software [32, 33].

## 2.9 Study management plan

**2.9.1 Trial management group.** The Trial Management Group including the Chief Investigator (Prof Simon Carding), the study research team, NHS Principal Investigators and Clinical Research Network's (CRN) Research Nurses will be responsible for the day-to-day management of the trial. They will monitor all aspects of the conduct and progress of the trial,

**Table 2. Power of the MOTION study to detect difference in cognitive function between groups defined by features of the microbiome.**

| Power to detect effect of feature on change in cognitive function (%) | Prevalence of feature (%) | | | | |
|---|---|---|---|---|---|
| Standardised effect size (standard deviations) | 50 | 25 | 10 | 7 | 5 |
| 0.22 | 50 | 39 | 22 | - | - |
| 0.36 | 90 | 77 | 47 | - | - |
| 0.45 | 99 | 97 | 67 | - | - |
| 0.68 | - | - | 99 | 87 | 50 |

ensure that the protocol is adhered to and take appropriate action to safeguard participants and the quality of the trial itself.

**2.9.2 Trial management oversight group.** Management of the study will be overseen by a Trial Management Oversight Group whose membership is made up of representatives from;

- Health, Safety, Environment and Quality Assurance.

- QIB Sponsor Representative.

- Gut Microbes and Health QIB Programme Manager.

- QIB Statistician.

- Information Technology Security Specialist.

- NRP Biorepository.

- Patient and Public Involvement.

- QIB Bioinformatician.

The Trial Management Oversight Group will be responsible for overseeing the running of the study. They will ensure the monitoring and facilitating the progress of the study, and ensure the study is delivered within the projected timelines. Recruitment targets, success of data collection, and any specific issues arising will be addressed.

## 2.10 Ethical and regulatory considerations

The MOTION study has been reviewed and agreed by the Human Research Governance Committee at the Quadram Institute Bioscience and the East Midlands—Nottingham 1 Research Ethics Committee (reference 19/EM/0055) and received written ethical approval by the Human Research Authority on the 1st April 2019. IRAS project ID number 241617.

The authors confirm that all ongoing and related trials for this drug/intervention are registered.

**2.10.1 Declaration of helsinki and relevant regulations.** The Investigator will ensure that this study is conducted in accordance with the principles of the Declaration of Helsinki. The proposed research will be conducted in accordance with the conditions and principles of the International Conference on Harmonisation Good Clinical Practice, and in compliance with national law. The research will meet the requirements of the new EU General Data Protection Regulation (GDPR), UK Data Protection Act 2018 and relevant sponsor's policies.

**2.10.2 Expenses and benefits.** Reasonable travel expenses for all study-related visits will be paid.

As a thank you to participants, each participant will receive a shopping voucher worth £10 at Study Visit 4 and a further voucher to the value £15 upon study completion at Study Visit 8.

It is not anticipated that any post-trial care will be required. If any participant is harmed whilst taking part in this clinical research study as a result of negligence on the part of a member of the study team, QIB holds liability insurance for such circumstances.

## 3 Discussion

There is a pressing social and economic need for research to promote health and independence into old age. The development of strategies that prevent or delay age-related disease and maintain lifelong health are a major goal over the coming decades. Identifying predictive biomarkers of conditions which affect this vulnerable group, such as dementia, is a key area of research needed so new treatments can be developed to prevent these conditions.

The MOTION study aims to provide key insights into how later-life microbiome profiles correlate with health outcomes, which will inform the development of new non-toxic, microbe-based therapies to promote and maintain health and/or treat disease.

Larger longitudinal studies, such as MOTION, with sequential (serial) sampling of the gut microbiome during ageing would provide a clearer picture of how the gut microbiome changes during ageing, and whether it is a contributing factor to declining health in old age and/or in the development of neurodegenerative disorders and dementia. A comprehensive longitudinal multiparameter-based study such as this is novel and has not been previously undertaken.

The major unanticipated operational issue impacting the MOTION study during 2020–2021 is the COVID-19 pandemic. Participant recruitment and testing was stalled or halted completely during the two UK national lockdowns. The study has, to date, been impeded by approximately 12 months in total when accounting for recruitment time lost and time required to catch up with missed appointments during lockdowns. We have received several withdrawals attributed to the pandemic and the additional stress brought on by these 'locked down' environments/ safety fears.

Several amendments to the study have been made to increase safety of participants and research staff during the pandemic, with the hope that these actions will mean the MOTION study will not be affected in the long term. It is likely COVID-19 will continue to have implications on future studies and risk assessments should be made prior to set up.

## Supporting information

**S1 File. Full MOTION study protocol approved by the ethics committee.**
(PDF)

**S2 File. SPIRIT checklist_MOTION.**
(PDF)

## Acknowledgments

We would like to thank the NRP Biorepository providing long term storage of samples from the MOTION study and for providing access to the Achiever Medical Laboratory Information Management System for the MOTION study team.

We would also like to thank the NNUH Patient Research Ambassadors Ron Brewer, Tony Jackson and Rosalinde Bailey for their involvement in the MOTION study Trial Management Oversight Group and their invaluable contributions and advice regarding participant involvement in this clinical research.

## Author Contributions

**Conceptualization:** Sarah Phillips, Antonietta Hayhoe, Michael Hornberger, Simon R. Carding.

**Data curation:** Rachel Watt, Thomas Atkinson, George M. Savva, Melissa Cambell-Kelly, Simon Rushbrook.

**Formal analysis:** George M. Savva, Michael Hornberger, Ben J. L. Burton, Janak Saada.

**Funding acquisition:** Simon R. Carding.

**Investigation:** Rachel Watt, Thomas Atkinson, Ben J. L. Burton, Melissa Cambell-Kelly, Simon Rushbrook.

**Methodology:** Rachel Watt, Thomas Atkinson, Shelina Rajan, George M. Savva, Michael Hornberger, Ben J. L. Burton, Janak Saada, Melissa Cambell-Kelly, Simon Rushbrook.

**Project administration:** Sarah Phillips, Rachel Watt, Shelina Rajan, Antonietta Hayhoe, Michael Hornberger, Ben J. L. Burton, Melissa Cambell-Kelly, Simon R. Carding.

**Supervision:** Sarah Phillips, Rachel Watt, Simon R. Carding.

**Writing – original draft:** Sarah Phillips.

**Writing – review & editing:** Simon R. Carding.

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
