## [Decision Letter · Decision Letter 0]

10 May 2022

PONE-D-21-18440MOTION Study Protocol – a longitudinal study in a cohort aged 60 years and older to obtain mechanistic knowledge of the role of the gut microbiome during normal healthy ageing and to develop strategies that will improve lifelong health and wellbeingPLOS ONE

Dear Dr. carding,

Thank you for submitting your manuscript to PLOS ONE. After careful consideration, we feel that it has merit but does not fully meet PLOS ONE’s publication criteria as it currently stands. Therefore, we invite you to submit a revised version of the manuscript that addresses the points raised during the review process.

The manuscript has been evaluated by two reviewers, and their comments are available below.

The reviewers’ comments raise some overlapping concerns about additional detail and clarification of the proposed methodology and statistical analyses, in addition to further discussion of the implications of the proposed work.

Could you please carefully revise the manuscript to address all comments raised?

We look forward to receiving your revised manuscript.

Kind regards,

Avanti Dey, PhD

Staff Editor

PLOS ONE

Journal Requirements:

“The primary sponsor of the project is the Quadram Institute Bioscience. The funders had no role in study design, data collection and analysis, decision to publish, or preparation of the manuscript.

The study has been funded by BBSRC through an Institute Strategic Programme (ISP) award to the QIB Gut Health and Food Safety Programme (BB/R012490/1), and its constituent projects BBS/E/F/000PR10353 and BBS/E/F/000PR10356. The study is adopted into the NIHR CRN Central Portfolio Management System (CPMS, add study number and speciality) portfolio which provides additional support in terms of hospital infrastructure and staff support. George Savva is funded through the BBSRC Core Capability Grant BB/CCG1860/1 at the Quadram Institute Bioscience. The Achiever Medical Laboratory Information Management System was procured using the BBSRC Capital Grant Award for the enhancement of the NRP Biorepository.”

We note that you have provided additional information within the Funding s Section that is not currently declared in your Funding Statement. Please note that funding information should not appear in the Acknowledgments section or other areas of your manuscript. We will only publish funding information present in the Funding Statement section of the online submission form.

“The study has been funded by the BBSRC through an Institute Strategic Programme (ISP) award to the QIB Gut Health and Food Safety Programme (BB/R012490/1) (SRC), and its constituent projects BBS/E/F/000PR10353 and BBS/E/F/000PR10356. The study is adopted into the NIHR CRN Central Portfolio Management System (CPMS, add study number and speciality) portfolio which provides additional support in terms of hospital infrastructure and staff support. GS is funded through the BBSRC Core Capability Grant BB/CCG1860/1 at the Quadram Institute Bioscience. The Achiever Medical Laboratory Information Management System was procured using the BBSRC Capital Grant Award for the enhancement of the Norwich Research Park Biorepository.”

“The study has been funded by the BBSRC through an Institute Strategic Programme (ISP) award to the QIB Gut Health and Food Safety Programme (BB/R012490/1) (SRC), and its constituent projects BBS/E/F/000PR10353 and BBS/E/F/000PR10356. The study is adopted into the NIHR CRN Central Portfolio Management System (CPMS, add study number and speciality) portfolio which provides additional support in terms of hospital infrastructure and staff support. GS is funded through the BBSRC Core Capability Grant BB/CCG1860/1 at the Quadram Institute Bioscience. The Achiever Medical Laboratory Information Management System was procured using the BBSRC Capital Grant Award for the enhancement of the Norwich Research Park Biorepository.”

Reviewers' comments:

Reviewer's Responses to Questions

**Comments to the Author**

1. Does the manuscript provide a valid rationale for the proposed study, with clearly identified and justified research questions?

Reviewer #1: Yes

Reviewer #3: Yes

2. Is the protocol technically sound and planned in a manner that will lead to a meaningful outcome and allow testing the stated hypotheses?

Reviewer #1: No

Reviewer #3: Yes

3. Is the methodology feasible and described in sufficient detail to allow the work to be replicable?

Reviewer #1: No

Reviewer #3: Yes

4. Have the authors described where all data underlying the findings will be made available when the study is complete?

Reviewer #1: Yes

Reviewer #3: Yes

5. Is the manuscript presented in an intelligible fashion and written in standard English?

Reviewer #1: Yes

Reviewer #3: Yes

6. Review Comments to the Author

You may also provide optional suggestions and comments to authors that they might find helpful in planning their study.

Reviewer #1: This paper is primarily a descriptive presentation of the protocol for the MOTION study which aims to determine whether and how changes in the gut microbiome are associated with physical and mental capacity. The authors consider this to be a comprehensive longitudinal multiparameter study. It is primarily correlational and section 3.8 outlines the sample size, power (Table 2) and analysis strategies for cross sectional analysis among the groups. Although the approach appears reasonable and the analyses anticipated is mixed effects models, the elaboration of the terms in the model and their random or fixed roles is not clear. The cohorts are not randomized and if any statistical comparison is anticipated, the investigators may wish to consider a propensity analysis approach.

Reviewer #3: The authors describe their study protocol for a longitudinal study in participants 60 years of age or older on the role of the gut microbiome during aging and cognitive function and well being. This is an interesting study. Appropriate methodology will be employed and the data will be of clinical significance. However, I have a few recommendations for consideration.

1. While the sample size is reasonable, I was wondering if a power calculation and sample size justification had been conducted.

2.Are the authors going to collect data on the sex and dietary patterns/practices over the course of the study.

3. How will the authors ensure compliance to the protocol?

4. The authors need to emphasize and elaborate on the novelty aspect of their proposition as well as the clinical applicability of the findings.

7. PLOS authors have the option to publish the peer review history of their article (what does this mean?). If published, this will include your full peer review and any attached files.

Reviewer #1: No

Reviewer #3: No

---

## [Author Response · Author response to Decision Letter 0]

17 Jun 2022

We would like to thank the reviewers for their careful and considered review of our manuscript. Below we have provided a point-by-point response, in blue text, to all of the queries raised in addition to a PDF copy as well as a Word version of the document with tracked changes. 

Journal Requirements:

The manuscript has been revised to meet these criteria.

The manuscript is a protocol paper, and the study has yet to complete the recruitment phase, and as such does not include any conclusions drawn and is yet to produce data which can be published.

“The primary sponsor of the project is the Quadram Institute Bioscience. The funders had no role in study design, data collection and analysis, decision to publish, or preparation of the manuscript.

The study has been funded by BBSRC through an Institute Strategic Programme (ISP) award to the QIB Gut Health and Food Safety Programme (BB/R012490/1), and its constituent projects BBS/E/F/000PR10353 and BBS/E/F/000PR10356. The study is adopted into the NIHR CRN Central Portfolio Management System (CPMS, add study number and speciality) portfolio which provides additional support in terms of hospital infrastructure and staff support. George Savva is funded through the BBSRC Core Capability Grant BB/CCG1860/1 at the Quadram Institute Bioscience. The Achiever Medical Laboratory Information Management System was procured using the BBSRC Capital Grant Award for the enhancement of the NRP Biorepository.”

We note that you have provided additional information within the Funding s Section that is not currently declared in your Funding Statement. Please note that funding information should not appear in the Acknowledgments section or other areas of your manuscript. We will only publish funding information present in the Funding Statement section of the online submission form.

“The study has been funded by the BBSRC through an Institute Strategic Programme (ISP) award to the QIB Gut Health and Food Safety Programme (BB/R012490/1) (SRC), and its constituent projects BBS/E/F/000PR10353 and BBS/E/F/000PR10356. The study is adopted into the NIHR CRN Central Portfolio Management System (CPMS, add study number and speciality) portfolio which provides additional support in terms of hospital infrastructure and staff support. GS is funded through the BBSRC Core Capability Grant BB/CCG1860/1 at the Quadram Institute Bioscience. The Achiever Medical Laboratory Information Management System was procured using the BBSRC Capital Grant Award for the enhancement of the Norwich Research Park Biorepository.”

We have removed the funding section from the manuscript and we have provided the following replacement text: 

The study was funded by the BBSRC through an Institute Strategic Programme (ISP) award to the QIB Gut Health and Food Safety Programme (BB/R012490/1), and its constituent projects BBS/E/F/000PR10353 and BBS/E/F/000PR10356. The study is adopted into the NIHR CRN Central Portfolio Management System (CPMS, add study number and speciality) portfolio which provides additional support in terms of hospital infrastructure and staff support. Dr George Savva is funded through the BBSRC Core Capability Grant BB/CCG1860/1 at the Quadram Institute Bioscience. The Achiever Medical Laboratory Information Management System was procured using the BBSRC Capital Grant Award for the enhancement of the NRP Biorepository.

“The study has been funded by the BBSRC through an Institute Strategic Programme (ISP) award to the QIB Gut Health and Food Safety Programme (BB/R012490/1) (SRC), and its constituent projects BBS/E/F/000PR10353 and BBS/E/F/000PR10356. The study is adopted into the NIHR CRN Central Portfolio Management System (CPMS, add study number and speciality) portfolio which provides additional support in terms of hospital infrastructure and staff support. GS is funded through the BBSRC Core Capability Grant BB/CCG1860/1 at the Quadram Institute Bioscience. The Achiever Medical Laboratory Information Management System was procured using the BBSRC Capital Grant Award for the enhancement of the Norwich Research Park Biorepository.”

Please see the response provided to 3. above.

We have actioned this in the revised manuscript.

Reviewers' comments:

Reviewer's Responses to Questions

Comments to the Author

1. Does the manuscript provide a valid rationale for the proposed study, with clearly identified and justified research questions?

Reviewer #1: Yes

Reviewer #3: Yes

2. Is the protocol technically sound and planned in a manner that will lead to a meaningful outcome and allow testing the stated hypotheses?

Reviewer #1: No

Reviewer #3: Yes

3. Is the methodology feasible and described in sufficient detail to allow the work to be replicable?

Reviewer #1: No

Reviewer #3: Yes

4. Have the authors described where all data underlying the findings will be made available when the study is complete?

Reviewer #1: Yes

Reviewer #3: Yes

5. Is the manuscript presented in an intelligible fashion and written in standard English?

Reviewer #1: Yes

Reviewer #3: Yes

6. Review Comments to the Author

You may also provide optional suggestions and comments to authors that they might find helpful in planning their study.

Reviewer #1: This paper is primarily a descriptive presentation of the protocol for the MOTION study which aims to determine whether and how changes in the gut microbiome are associated with physical and mental capacity. The authors consider this to be a comprehensive longitudinal multiparameter study. It is primarily correlational, and section 3.8 outlines the sample size, power (Table 2) and analysis strategies for cross sectional analysis among the groups. Although the approach appears reasonable and the analyses anticipated is mixed effects models, the elaboration of the terms in the model and their random or fixed roles is not clear. The cohorts are not randomized and if any statistical comparison is anticipated, the investigators may wish to consider a propensity analysis approach.

Thank you for your comment. As you rightly point out the data is not randomised and controlling for potentially confounding variables and selection bias will be important in our analysis. We are yet to prepare statistical analysis plans to address the main questions underlying the report, the study protocol does not specify any particular analysis and methods to statistically analyse microbiome data are ongoing rapid development, but we will carefully consider ‘epidemiological’ concerns and potential approaches such as the use of propensity scores when we are developing these. According to our study protocol analysis plans will be developed and pre-registered prior to analysis.

The mixed models underlying the power calculations were necessary simple and only provide a very indicative guide for general estimation of the effect of dichotomous microbiome features on cognitive change. We have slightly expanded the text to specify that there were fixed effects of the interaction between the microbiome, time and their interaction and a random slope and intercept for cognitive function for each participant. Time was considered a linear continuous variable in the analysis. 

We appreciate there are many assumptions underlying power calculations from mixed effects models, not least the fact that the minimum effect size and variance were based on a cognitive test we did not include in our study (hence our use of unitless standardised effect sizes in the table); but despite the assumptions underlying this model, extremely similar values for power can be derived by considering t-tests of simple comparison of change scores over time. 

For example, if we consider the effect size of 0.36 standard deviations between groups, then a sample size of 300 and a prevalence for the feature of interest of 25% leads to a power of 40%, similar to the 39% quoted from the simulation (using the pwr::pwr.t2n.test package for R). With a prevalence of 50%, the power is 87% compared to the 90% quoted from our simulation. While this does not correspond exactly to the analyses that we will conduct, both this simulation and this analytic approach provide some indication of the range of effect sizes that will be detectable.Reviewer #3: The authors describe their study protocol for a longitudinal study in participants 60 years of age or older on the role of the gut microbiome during aging and cognitive function and well being. This is an interesting study. Appropriate methodology will be employed and the data will be of clinical significance. However, I have a few recommendations for consideration.

1. While the sample size is reasonable, I was wondering if a power calculation and sample size justification had been conducted. 

The discussion of sample size is found in section 3.8.4.

2.Are the authors going to collect data on the sex and dietary patterns/practices over the course of the study. 

The health questionnaires that participants complete includes questions include their sex and information on daily food intake. It also asks what type of diet they usually follow (e.g. plant/meat based or if they follow any special diets) and if they have any conditions such as gluten/ lactose intolerances or allergies. We have amended the manuscript to give more detail on what is included in this questionnaire.

3. How will the authors ensure compliance to the protocol? 

The Trial Management Group including the Chief Investigator (Prof Simon Carding), the study research team, NHS Principal Investigators and Clinical Research Network’s (CRN) Research Nurses will be responsible for the day-to-day management of the trial. They will monitor all aspects of the conduct and progress of the trial, ensure that the protocol is adhered to and take appropriate action to safeguard participants and the quality of the trial itself. This is noted in the study management plan within the manuscript.

4. The authors need to emphasize and elaborate on the novelty aspect of their proposition as well as the clinical applicability of the findings. 

Thank you for your comment. We have reviewed the introduction and have added an additional statement to highlight the novelty of the MOTION study.

---

## [Decision Letter · Decision Letter 1]

2 Aug 2022

PONE-D-21-18440R1MOTION Study Protocol – a longitudinal study in a cohort aged 60 years and older to obtain mechanistic knowledge of the role of the gut microbiome during normal healthy ageing and to develop strategies that will improve lifelong health and wellbeingPLOS ONE

Dear Dr. carding,

Thank you for submitting your manuscript to PLOS ONE. After careful consideration, we feel that it has merit but does not fully meet PLOS ONE’s publication criteria as it currently stands. Therefore, we invite you to submit a revised version of the manuscript that addresses the points raised during the review process. One of the reviewers raises some minor outstanding points, can you please address these in a further revision?

We look forward to receiving your revised manuscript.

Kind regards,

Avanti Dey, PhD

Staff Editor

PLOS ONE

Journal Requirements:

Reviewers' comments:

Reviewer's Responses to Questions

**Comments to the Author**

1. Does the manuscript provide a valid rationale for the proposed study, with clearly identified and justified research questions?

Reviewer #1: Yes

Reviewer #3: Yes

2. Is the protocol technically sound and planned in a manner that will lead to a meaningful outcome and allow testing the stated hypotheses?

Reviewer #1: Partly

Reviewer #3: Yes

3. Is the methodology feasible and described in sufficient detail to allow the work to be replicable?

Reviewer #1: Yes

Reviewer #3: Yes

4. Have the authors described where all data underlying the findings will be made available when the study is complete?

Reviewer #1: Yes

Reviewer #3: Yes

5. Is the manuscript presented in an intelligible fashion and written in standard English?

Reviewer #1: Yes

Reviewer #3: Yes

6. Review Comments to the Author

You may also provide optional suggestions and comments to authors that they might find helpful in planning their study.

Reviewer #1: The authors defend the mixed model approach in their responses as well as the use of random and fixed terms.

However, this reviewer does not understand the comment ,"We are yet to prepare statistical analysis plans to address the main questions underlying the report, the study protocol does not specify any particular analysis and methods to statistically analyse microbiome data are ongoing rapid development, but we will carefully consider ‘epidemiological’ concerns and potential approaches such as the use of propensity scores when we are developing these. According to our study protocol analysis plans will be developed and pre-registered prior to analysis. "

Section 3.8 and Table 2 appear to address this concern. This is confusing.

Reviewer #3: The authors have addressed all initial concerns and have adequately revised their manuscript. I have no further comments.

7. PLOS authors have the option to publish the peer review history of their article (what does this mean?). If published, this will include your full peer review and any attached files.

Reviewer #1: No

Reviewer #3: No

---

## [Author Response · Author response to Decision Letter 1]

25 Aug 2022

Reviewer #1: The authors defend the mixed model approach in their responses as well as the use of random and fixed terms.

However, this reviewer does not understand the comment ,"We are yet to prepare statistical analysis plans to address the main questions underlying the report, the study protocol does not specify any particular analysis and methods to statistically analyse microbiome data are ongoing rapid development, but we will carefully consider ‘epidemiological’ concerns and potential approaches such as the use of propensity scores when we are developing these. According to our study protocol analysis plans will be developed and pre-registered prior to analysis. "

Section 3.8 and Table 2 appear to address this concern. This is confusing.

We apologise for the confusion. The analysis section of our paper outlines a number of research questions that will be answered as opposed to the specific statistical methods to address them. For the sake of a power calculation, we used a very simplified analysis linking a single binary microbiome feature (for example the baseline presence of a particular taxon) to cognitive change, but analysis of many features of the microbiome are more difficult to analyse than this being compositional, multivariate, sparse, based on counts etc. Methods to measure microbiomes and to analyse microbiome data are evolving, particularly with respect to more complex analysis (e.g., for longitudinal data) and so we have not yet finalised the statistical analysis methods for these.

---

## [Decision Letter · Decision Letter 2]

29 Sep 2022

MOTION Study Protocol – a longitudinal study in a cohort aged 60 years and older to obtain mechanistic knowledge of the role of the gut microbiome during normal healthy ageing and to develop strategies that will improve lifelong health and wellbeing

PONE-D-21-18440R2

Dear Dr. carding,

Thank you for the most recent submission of your protocol paper that has been passed on to me. It is clearly an important study of significant potential interest and I am sorry for the rather drawn out review process. We are now pleased to inform you that your manuscript has been judged scientifically suitable for publication and will be formally accepted for publication once it meets all outstanding technical requirements.

Kind regards,

Antony Bayer

Academic Editor

PLOS ONE

Additional Editor Comments (optional):

Reviewers' comments:

Reviewer's Responses to Questions

**Comments to the Author**

1. Does the manuscript provide a valid rationale for the proposed study, with clearly identified and justified research questions?

Reviewer #3: Yes

2. Is the protocol technically sound and planned in a manner that will lead to a meaningful outcome and allow testing the stated hypotheses?

Reviewer #3: Yes

3. Is the methodology feasible and described in sufficient detail to allow the work to be replicable?

Reviewer #3: Yes

4. Have the authors described where all data underlying the findings will be made available when the study is complete?

Reviewer #3: Yes

5. Is the manuscript presented in an intelligible fashion and written in standard English?

Reviewer #3: Yes

6. Review Comments to the Author

You may also provide optional suggestions and comments to authors that they might find helpful in planning their study.

Reviewer #3: The authors have addressed all initial concerns and have adequately revised their manuscript. I am satisfied with the revised manuscript. I have no further comments.

7. PLOS authors have the option to publish the peer review history of their article (what does this mean?). If published, this will include your full peer review and any attached files.

Reviewer #3: No

---

## [Editor Report · Acceptance letter]

27 Oct 2022

PONE-D-21-18440R2 

A protocol paper for the MOTION Study – a longitudinal study in a cohort aged 60 years and older to obtain mechanistic knowledge of the role of the gut microbiome during normal healthy ageing in order to develop strategies 

Dear Dr. Carding:

I'm pleased to inform you that your manuscript has been deemed suitable for publication in PLOS ONE. Congratulations! Your manuscript is now with our production department. 

Kind regards, 

on behalf of

Professor Antony Bayer 

Academic Editor

PLOS ONE